# How Far Can LLMs Go on Cognitive Health Prediction? A Study on EMA Data

**Sambit Mukherjee, ORCID**[1,†]    **Anthony Fu, ORCID**[1,2,†]    **Kai Zhao, ORCID**[3]    **Yang Ren, ORCID**[4]
**Dezhi Wu**, ORCID[1]    **Chih-Hsiang Jason Yang**, ORCID [1]

[1]University of South Carolina    [2]Dutch Fork High School    [3]Workday AI    [4]Yale University
[†] Equal Contribution

## Abstract

Cognitive health is a growing public-health burden in an increasingly aging world, and scalable methods for monitoring cognitive state are urgently needed. Smartphone-based intensive measurement studies can capture short-term cognitive variability in daily life via repeated ecological momentary assessment (EMA) surveys and brief cognitive micro-tests, but the resulting datasets are small, heterogeneous, and predominantly tabular, raising a basic question: can large language models (LLMs) serve as effective predictors in this structured regime, or do specialized tabular learners remain necessary? We study a proprietary healthy aging EMA dataset of 115 older adults (ages 65–91) collected over a two-week measurement burst, and predict an ordinal cognitive severity septile derived from a composite cognition score using participant-level splits to evaluate generalization to unseen individuals. We benchmark state-of-the-art LLMs under a unified, machine-parseable structured-output interface against strong tabular machine-learning and statistical ordinal baselines using exact accuracy, tolerance accuracy, and macro-F1. Across models and settings, tree-based tabular methods achieve the best boundary-accurate performance, while LLMs capture coarse ordinal ordering but are less reliable at exact septile decisions; moreover, structured-output failures emerge as a practical deployment constraint that must be included in end-to-end evaluation. Our results delineate the current limits of prompting-based LLM inference for structured ambulatory cognition prediction and provide a reproducible framework for comparing LLMs to tabular baselines in small-sample clinical monitoring tasks.

## 1 Introduction

Cognitive health is becoming a defining challenge of an aging world. As the proportion of older adults grows, even modest changes in attention, memory, and executive function can translate into meaningful differences in independence, safety, and quality of life. Clinically, timely identification of worsening cognitive states matters because it can trigger earlier support, more tailored care, and closer monitoring for individuals at elevated risk. Yet cognition in daily life is not static: it fluctuates with sleep, stress, mood, pain, and activity. Traditional clinic-based assessments, while valuable, often provide only sparse snapshots and can miss short-term variability that may be informative about an individual's current condition.

Smartphone-based intensive measurement studies address this gap by collecting ecological momentary assessment (EMA) self-reports together with brief cognitive micro-tests repeatedly over short measurement bursts (Rast et al., 2012; Brewster et al., 2021). These datasets are particularly important because they capture cognition where it happens—in the flow of daily life—rather than only in the clinic. At the same time, they create a modeling setting that is both practically relevant and technically challenging: sample sizes are modest, adherence varies, missingness is common, and features are heterogeneous, mixing survey items, behavioral summaries, and task latency signals.

---

[†]Co-first authors made equal contributions. Contact emails: Sambit Mukherjee: SAMBIT@email.sc.edu & Anthony Fu: anthonyzhfu@gmail.com.

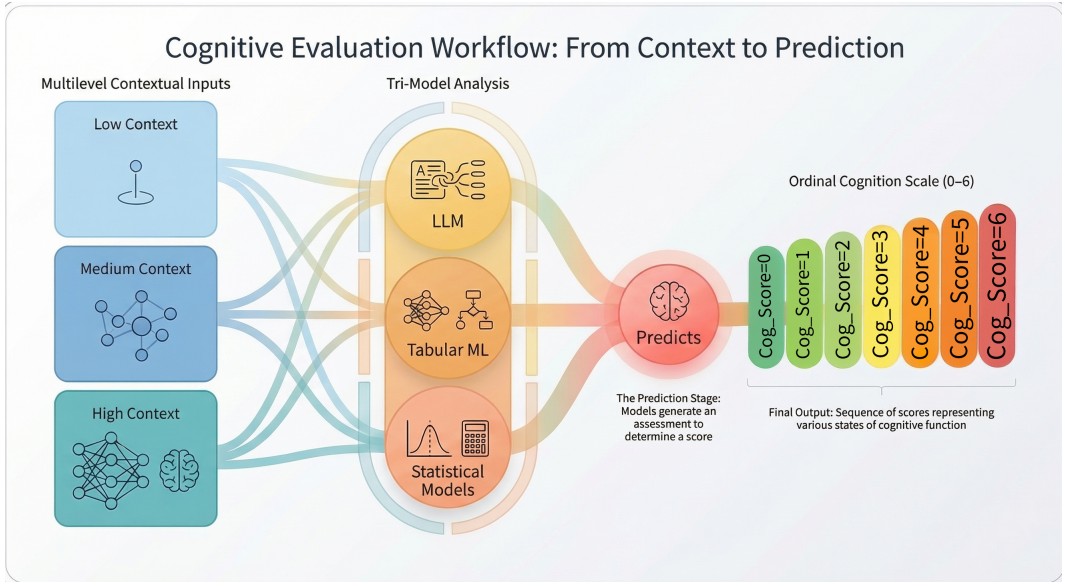

Figure 1: Overview of our evaluation. Low, medium, and high contexts for cognitive measurement inputs are provided to LLM, tabular ML, and statistical models to predict a ordinal Cognition Score(Cog_Score): 0–6.

A key research gap is that it remains unclear: how well large language models (LLMs) perform on this kind of structured, tabular cognition prediction, and how their performance compares to strong, purpose-built tabular machine learning (ML) models and classical statistical ordinal baselines. Although LLMs can express plausible clinical reasoning in text, in many real deployments they are used via prompting and in-context learning rather than weight updating, which may limit their ability to learn sharp numeric decision boundaries from structured features. This gap matters because practitioners increasingly face a concrete choice: use an LLM as a general inference engine, or rely on specialized tabular predictors that are historically strong in small-data settings.

In this paper, we study this question on a proprietary EMA healthy aging dataset collected from 115 older adults (65 to 91 years of age) at risk of neuropsychological disease but without dementia at enrollment (See Figure 1) in three levels of contexts with different features, including low context (i.e., sex, age, marriage, count of steps, sleep quality, duration of sleep), medium context (i.e., features of low context + physical, feel, joyful, calm, feel energy, energetic) and high context (i.e., features of medium context + present, control, anxious, stressed, tired, BMI, symbol_search accuracy, dot_memory trials, go_nogo trails). More details on the context levels are illustrated in Figure 2. During a two-week measurement burst, participants completed repeated EMA surveys along with NIH-funded smartphone cognitive tasks, including symbol search, dot memory, and go/no-go assessments. These repeated observations yield a composite cognition score that summarizes the subjective cognitive state in daily life; we discretize this score into an ordinal septile severity label and treat the problem as a structured ordinal prediction based on mixed survey, behavioral, and task-derived features. To ensure that the evaluation reflects real-world generalization, we use participant-level train/validation/test splits so that test assessments come from individuals never seen during training.

Our empirical study is designed to make comparisons between modeling families as fair and practically meaningful as possible. We benchmark several state-of-the-art LLMs under a unified prompting interface that constrains each prediction to a single-line, machine-parseable JSON output, enabling deterministic scoring and failure auditing. In parallel, we train strong tabular ML baselines that represent the default toolkit for small-sample structured prediction in health settings, and we include interpretable statistical ordinal models to provide coefficient-based comparators that explicitly exploit label ordering. We evaluate models using exact septile accuracy as the primary metric, complemented by tolerance-based accuracy (within $\pm 1$ or $\pm 2$ septiles) and macro-F1 to reflect class imbalance and ordinal near-miss behavior.

Across metrics and experimental configurations, the results reveal a stable pattern that clarifies the practical role of LLMs in structured cognition modeling. Tree-based tabular models deliver the highest exact septile accuracy and macro-F1, reflecting their advantage in learning sharp decision boundaries in heterogeneous numeric feature spaces. LLMs, in contrast, frequently capture the coarse ordering of severity—achieving substantially higher tolerance accuracy than exact accuracy—but are less reliable at placing predictions on the correct septile boundary, which is the core requirement for precise ordinal classification. Beyond accuracy, we find that operational reliability is itself a first-order consideration: structured-output parse failures occur for some LLMs and must be accounted for as part of end-to-end system performance rather than discarded as incidental noise.

Our paper has following contributions:

- We provide a focused benchmark for ambulatory cognition severity prediction on a real longitudinal healthy aging EMA dataset, with a clearly specified ordinal target construction and participant-level splitting that tests generalization to unseen individuals.

- We present a head-to-head evaluation of LLM prompting against strong tabular ML and statistical ordinal baselines under a unified metric suite that includes both exact and tolerance-based accuracy, making it possible to distinguish "near-miss" ordinal reasoning from true boundary-accurate prediction.

- We explicitly treat structured-output compliance as part of model behavior by reporting parse failures and emphasizing that reliability constraints are necessary for deployable LLM-based clinical pipelines, particularly in settings where predictions must be auditable and automatically consumable.

## 2 EXPERIMENTAL SETTING

### 2.1 DATA SETS

We evaluate on a proprietary healthy aging EMA study conducted by *ECHO Lab* at *TecHealth Center, University of South Carolina*. The dataset contains 115 older adults (ages 65–91) with neuropsychological risk factors but without dementia diagnoses at enrollment. Each participant completed a two-week measurement burst in a controlled protocol. Cognitive micro-tests were administered via the NIH-funded M2C2 smartphone app and included a symbol search task, a grid-based dot memory task, and a go/no-go continuous performance task, each providing reaction-time and/or accuracy-derived metrics. In parallel, EMA surveys were delivered four semi-random times per day during waking hours to capture subjective cognition and affective state, including items such as mindsharpness, concentration, anxiety, stress, happiness, and related constructs. A composite cognition score cog in the range 0–100 is derived from these self-reports, and we discretize it into an ordinal septile label css in $\{0,\ldots,6\}$, where 0 corresponds to the poorest cognitive function and 6 to the highest. Importantly, the EMA, activity, and smartphone task measures were collected for research monitoring rather than clinical diagnosis; they are intended for within- and between-individual comparisons of short-term cognitive state and variability, and should not be interpreted as diagnostic evidence of dementia. All participants were recruited as older adults with elevated risk factors for dementia and did not have a dementia diagnosis at enrollment.

The preprocessed feature set used in our experiments consists of 25 variables drawn from three sources: EMA cognitive and affective items, objective task performance metrics, and a compact set of demographic/behavioral covariates (e.g., sex, marital status, BMI, age, daily steps, sleep quality, and sleep duration). Missing values are normalized from common string patterns (e.g., NA variants) into standard missing indicators; for LLM prompting, we represent missing or unknown values explicitly as -1 to reduce ambiguity and to keep prompts parseable. We split the dataset at the participant level to ensure that test evaluations reflect generalization to unseen individuals. The resulting partitions contain 70% training assessments (n=1011), 10% validation assessments (n=154), and 20% test assessments (n=296), with stratification to preserve the ordinal label distribution.

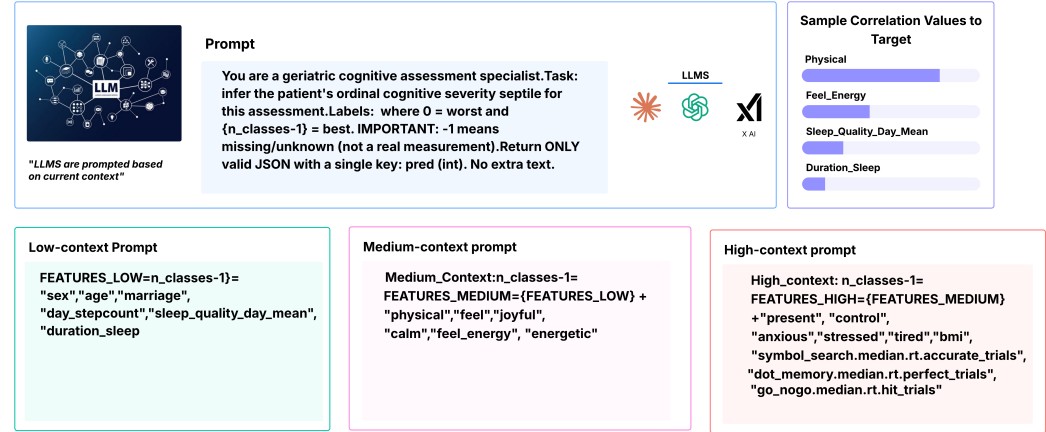

Figure 2: Standardized LLM prompt template and context-dependent feature sets. A single instruction template (ordinal severity prediction; JSON-only response; missing values encoded as −1) is paired with three nested context levels that systematically vary from demographics/behavior to affective/emotional and task-performance measures.

## 2.2 MODELS

### 2.2.1 LLM MODELS

In this study, we evaluated three state-of-the-art LLMs to examine their ability to predict cognitive severity. We specifically used Claude Opus 4.5, GPT 5.2, and Grok 4-Fast. All LLMs were tasked with inferring the patient's cognitive severity septile (CSS) score ranging from 0 to 6, based on demographic and survey-derived features. To ensure results were comparable across models, we used the same minimal prompts for all three LLMs. The prompt was intentionally kept barebones to isolate the effect that different contextual levels have on model reasoning, yielding more distinct results rather than relying on heavy prompt engineering, thereby allowing the model's intrinsic capabilities to really shine.

Three advanced LLMs were assessed to investigate their capacity in predicting cognitive severity. Specifically, Claude Opus 4.5, GPT 5.2, and Grok 4-Fast were employed. Each LLM was assigned the task of predicting the patient's cognitive severity septile (CSS) score within a range of 0 to 6, utilizing demographic details and survey-based features. As seen in **Figure 2**, to ensure a uniform comparison among the models, identical minimal prompts were utilized for all three LLMs. The intentional simplicity of the prompts aimed to isolate the impact of varying contextual depths on model inference, thereby generating more distinct outcomes without heavy reliance on intricate prompt construction, thus enabling the intrinsic capabilities of the models to be effectively showcased.

### 2.2.2 ML MODELS

To establish standardized benchmarks for predicting CSS, four conventional Machine Learning Models were used to learn ordinal cognitive trends from quantitative and demographic attributes. The training encompassed three prevalent machine learning techniques: XGBoost regression, CatBoost regression, Extra Trees, and FTTTransformer. Each model was trained to generate a continuous assessment of cognitive intensity aligned with one of the six CSS tiers (0-6) for evaluation purposes. Conventional machine learning methods remain essential benchmarks for several reasons. Firstly, interpretable models such as ExtraTrees reveal the influential features driving predictions, a crucial aspect for their acceptance in clinical settings (Burkart & Huber, 2021). Secondly, gradient boosting techniques such as XGBoost and Catboost have historically demonstrated the ability to achieve accuracy comparable to that of traditional models using smaller training datasets (Shaffi et al., 2024). The FTTokenizer was selected as the designated deep learning

model for this study to assess its efficacy in processing a combination of diverse data types, including ordinal scores (e.g., mindsharp, concentrate) and behavioral measurements (e.g., step count, sleep quality). Unlike conventional neural networks that require substantial preprocessing efforts, FTTransformers are recognized for their ability to effectively manage data heterogeneity via learned tokenization strategies (Zhou & Chen, 2025).

### 2.2.3 STATISTICAL MODELS

To enhance machine-learning baselines with interpretable comparators that exhibit parametric characteristics, four classical statistical classifiers, including OrdinalLogitIT(L2), OrdinalLogitAT(L2), MultinomLogReg, and LDA, were assessed. The primary functions of these models within this research are twofold: 1) To offer transparent baseline models based on coefficients that are more straightforward for clinical auditing compared to intricate ensembles. In healthcare applications, interpretability is paramount for clinical adoption and decision-making. In the realm of healthcare utilization, achieving interpretability is crucial for acceptance in clinical settings and to facilitate decision-making processes (Pant et al., 2025), and our methodology enables a deliberate examination of the impact of integrating the inherent hierarchy of cognitive severity designations on the efficacy of the model in contrast to models that view categories as non-hierarchical. Research indicates that ordinal regression techniques derive considerable advantage from the utilization of hierarchical data, resulting in enhanced precision and increased fidelity of forecasts aligned with authentic ordinal values (Gutiérrez et al., 2016). The models chosen for the study encompassed OrdinalLogitIT (L2) and OrdinalLogitAT (L2) due to the requirement of handling an ordered septile outcome range (0–6), which required models that explicitly account for label ordering (Vargas et al., 2019). Ordinal logistic models based on thresholds, such as those using cumulative link functions, employ ordered decision thresholds to capture the underlying structure of ordinal outcomes. Multinomial Logistic Regression was used as a nominal-class reference point, treating cognitive severity classes as nominal categories without accounting for their inherent ordering. Although standard nominal classification methods can be applied to ordinal regression, there are algorithms that can better exploit the ordering information. Additionally, LDA was included as a classical statistical benchmark that assumes distinct feature distributions while offering interpretability. Widely used as a baseline classifier in medical contexts for multi-class scenarios (Li et al., 2019), LDA offers an alternative perspective on feature relevance through its discriminant functions.

## 3 EXPERIMENTAL RESULTS

### 3.1 MAIN RESULTS

**Table 1** summarizes test-set performance (296 held-out assessments from unseen participants) across LLM, tabular ML, and statistical baselines. Overall, tree-based tabular models provide the strongest boundary-accurate cognition severity predictions. ExtraTrees achieves the best performance across all methods with ToleranceAccuracy $\pm 0$ (TA@0) $= 0.57$ and Macro-F1 $= 0.55$, followed by XGBoost with ToleranceAccuracy $\pm 0$ (TA@0) $= 0.52$ and Macro-F1 $= 0.50$. CatBoost is comparable (TA@0 $= 0.50$), while the deep tabular baseline (FTTTransformer) underperforms tree ensembles on exact accuracy (TA@0 $= 0.42$) and Macro-F1 (0.39), despite high tolerance accuracy.

Among LLMs, medium-context prompting consistently yields the best results for all three models in this table. GPT 5.2 is the strongest LLM at medium context with ToleranceAccuracy $\pm 0$ (TA@0) $= 0.32$ and Macro-F1 $= 0.29$, narrowly ahead of Grok-4-fast at medium context (TA@0 $= 0.30$, Macro-F1 $= 0.27$) and Claude-Opus-4.5 at medium context (TA@0 $= 0.21$, Macro-F1 $= 0.23$). A salient pattern is that LLMs achieve substantially higher tolerance accuracy than exact accuracy: for example, GPT 5.2 improves from TA@0 $= 0.32$ to TA@0 $= 0.32$, ToleranceAccuracy $\pm 1$ (TA@1) $= 0.75$, and ToleranceAccuracy $\pm 2$ (TA@2) $= 0.92$, indicating that many LLM errors are near-misses in ordinal distance rather than large deviations. However, this coarse ordering does not translate into boundary-accurate septile decisions, which remain the primary requirement for precise ordinal classification.

Table 1: Test-set performance on 296 held-out evaluations ($N_{\text{eval}} = 296$) comparing LLMs (best-shot configurations at low, medium, and high context prompting), tabular ML models, and statistical baselines. Reported metrics include Tolerance Accuracy within $\pm 0$ (TA@0), $\pm 1$ (TA@1), and $\pm 2$ (TA@2), classification performance via Macro-F1, and error metrics RMSE (Root Mean Squared Error) and MAE (Mean Absolute Error).

| Group | Model | Ctx | TA@0 | TA@1 | TA@2 | RMSE | MAE | Macro-F1 | $n_{\text{eval}}$ | Shots |
|---|---|---|---|---|---|---|---|---|---|---|
| LLM | Claude-Opus-4.5 | L | 0.14 | 0.45 | 0.66 | 2.35 | 1.91 | 0.12 | 296 | 12 |
| | | M | 0.21 | 0.77 | 0.97 | 1.28 | 1.04 | 0.23 | 296 | 12 |
| | | H | 0.14 | 0.36 | 0.65 | 2.28 | 1.93 | 0.13 | 296 | 12 |
| | GPT 5.2 | L | 0.11 | 0.39 | 0.62 | 2.66 | 2.18 | 0.10 | 296 | 12 |
| | | M | 0.32 | 0.75 | 0.92 | 1.42 | 1.03 | 0.29 | 296 | 12 |
| | | H | 0.23 | 0.63 | 0.87 | 1.69 | 1.31 | 0.23 | 296 | 12 |
| | Grok-4-fast | L | 0.17 | 0.44 | 0.67 | 2.55 | 2.01 | 0.16 | 296 | 8 |
| | | M | 0.30 | 0.70 | 0.90 | 1.55 | 1.13 | 0.27 | 296 | 8 |
| | | H | 0.18 | 0.46 | 0.69 | 2.32 | 1.84 | 0.18 | 296 | 8 |
| ML | XGBoost | N/A | 0.52 | 0.91 | 0.99 | 0.90 | 0.57 | 0.50 | 296 | N/A |
| | CATBoost | N/A | 0.50 | 0.90 | 0.98 | 0.97 | 0.62 | 0.47 | 296 | N/A |
| | ExtraTrees | N/A | 0.57 | 0.92 | 0.99 | 0.91 | 0.54 | 0.55 | 296 | N/A |
| | FTTTransformer | N/A | 0.42 | 0.91 | 0.97 | 1.02 | 0.70 | 0.39 | 296 | N/A |
| Statistical | OrdinalLogit_IT (L2) | N/A | 0.48 | 0.91 | 0.97 | 0.95 | 0.62 | 0.46 | 296 | N/A |
| | OrdinalLogit_AT (L2) | N/A | 0.50 | 0.90 | 0.98 | 0.93 | 0.61 | 0.47 | 296 | N/A |
| | MultinomLogReg | N/A | 0.50 | 0.90 | 0.99 | 0.91 | 0.60 | 0.46 | 296 | N/A |
| | LDA | N/A | 0.44 | 0.88 | 0.99 | 0.96 | 0.67 | 0.40 | 296 | N/A |

Table 2: Validation Performance across LLM baselines for (0/7/8/12) shots for medium context.

| Group | Model | Shots | TA@0 | TA@1 | TA@2 | RMSE | MAE(Mean Absolute Error) | Macro-F1 | $n_{\text{eval}}$ |
|---|---|---|---|---|---|---|---|---|---|
| LLM | Claude-Opus-4.5 | 0 | 0.19 | 0.41 | 0.63 | 2.16 | 1.80 | 0.15 | 154 |
| | | 7 | 0.43 | 098 | 1 | 0.79 | 0.59 | 0.47 | 154 |
| | | 8 | 0.25 | 0.83 | 1 | 1.11 | 0.91 | 0.24 | 154 |
| | | 12 | 0.37 | 0.90 | 1 | 0.95 | 0.72 | 0.36 | 154 |
| | GPT 5.2 | 0 | 0.09 | 0.24 | 0.39 | 2.97 | 2.64 | 0.04 | 154 |
| | | 7 | 0.35 | 0.90 | 1 | 0.96 | 0.74 | 0.38 | 154 |
| | | 8 | 0.33 | 0.93 | 1 | 0.92 | 0.73 | 0.38 | 154 |
| | | 12 | 0.5 | 0.96 | 1 | 0.77 | 0.53 | 0.54 | 154 |
| | Grok-4-fast | 0 | 0.12 | 0.33 | 0.67 | 2.29 | 1.97 | 0.09 | 154 |
| | | 7 | 0.38 | 0.74 | 0.96 | 1.32 | 0.94 | 0.36 | 154 |
| | | 8 | 0.35 | 0.76 | 0.93 | 1.34 | 0.96 | 0.33 | 154 |
| | | 12 | 0.32 | 0.33 | 0.67 | 2.29 | 1.97 | 0.09 | 154 |

Classical statistical baselines are competitive with, and in several cases exceed, the strongest LLM configurations on exact accuracy. OrdinalLogit_AT and MultinomLogReg both reach TA@0 = 0.50 with Macro-F1 around 0.46–0.47, while OrdinalLogit_IT attains TA@0 = 0.48. These results suggest that a substantial fraction of predictive signal is approximately monotone/linear and can be captured by transparent models that explicitly exploit label ordering, but that tree ensembles provide additional gains likely through nonlinear interactions and threshold effects. Finally, the context sensitivity of LLMs is pronounced: moving from medium to high context reduces exact accuracy for all LLMs in Table 1 (e.g., GPT 5.2 drops from TA@0 = 0.32 to 0.23), supporting the conclusion that more context is not consistently beneficial for structured ordinal prediction and may introduce noise that distracts from the underlying numeric decision boundaries.

## 3.2 LESS CONTEXT VS. MORE CONTEXTS

We evaluate three prompt context levels (low, medium, high) to test the common assumption that adding more participant information improves LLM cognitive health prediction. Table 1 shows a consistent pattern across models: medium context yields the best performance, while high context degrades accuracy relative to medium. For GPT 5.2, moving from low to medium context improves exact accuracy from TA@0 = 0.11 to 0.32 and reduces the Root Mean Squared Error(RMSE) from $2.66 \to 1.42$, but adding further information in the high-context condition drops performance (TA@0 = 0.23, RMSE 1.69). Claude-Opus-4.5 and Grok-4-fast exhibit the same trend, peaking at

medium context and worsening under high context. This "medium-is-best" behavior suggests that additional information is not uniformly helpful for structured ordinal prediction and can introduce distraction or noise, making boundary placement less stable, aligning with observations that context sufficiency matters in clinical decision support settings. (Antoniadi et al., 2021).

### 3.3 Zero-shot vs. Few-shot Prompting

We use zero-shot prompting as a calibration-free baseline and few-shot prompting to test whether a small number of labeled examples improves ordinal boundary decisions. Prior work shows that LLMs can operate in clinical domains without task-specific training, particularly under data scarcity (Naliyatthaliyazchayil et al., 2025), and prompting strategies can materially affect stability and accuracy (Sivarajkumar et al., 2024). In our setting, zero-shot prompts evaluate whether the model can map structured EMA/task features to an ordinal severity scale using only the task description. Few-shot prompting, in contrast, provides a small set of labeled examples (7, 8, or 12 shots) drawn from training participants, which supplies direct evidence about this dataset's label semantics and feature conventions. This is especially important here because the target is discretized and ordinal, so a large fraction of error comes from misplacing thresholds rather than misunderstanding the general concept of "worse" versus "better."

Across models, few-shot prompting produces substantially stronger and more reliable performance than zero-shot, with the best configurations using 12 shots for GPT 5.2 and Claude-Opus-4.5 and 8 shots for Grok-4-fast (**Table. 2**). These gains are best understood as calibration rather than knowledge acquisition: the labeled examples help the model internalize the dataset-specific mapping from mixed features to septile labels, interpret the missing-value convention, and adhere to the strict structured-output format. Without this calibration, zero-shot predictions are more sensitive to generic prompt biases and may over-prefer certain labels or overweight superficial cues, leading to weaker exact accuracy and higher error. In short, for ordinal cognition prediction from structured features, zero-shot is useful as a sanity check for out-of-the-box behavior, but few-shot prompting is necessary for consistent boundary placement and stable evaluation.

## 4 Related Works

Smartphone-based intensive measurement designs, combining ecological momentary assessment (EMA) with brief cognitive micro-tests, have become a practical way to characterize cognition in daily life with high ecological validity and reduced recall bias (Shiffman et al., 2008; Stone & Shiffman, 1994). Prior work shows that unsupervised smartphone cognitive assessments can be feasible and psychometrically meaningful in older adults, including at-risk cohorts, supporting the premise that ambulatory measures capture clinically relevant differences while preserving context-dependent variability (Nicosia et al., 2023). A complementary theme in this literature is that short-term intraindividual variability (IIV), especially in reaction-time measures, can carry information beyond mean performance; longitudinal evidence links reaction-time IIV to cognitive impairment, dementia-related outcomes, and mortality risk (Haynes et al., 2017).

In parallel, behavioral and lifestyle signals—most notably physical activity—are repeatedly associated with cognitive health, motivating the inclusion of activity and related covariates when modeling cognitive state (Iso-Markku et al., 2022), with measurement validity often discussed through established wearable instrumentation such as activPAL3 (Sellers et al., 2016). Methodologically, these ambulatory datasets tend to be small, heterogeneous, and predominantly tabular after aggregation, where tree-based methods and boosting remain strong defaults for capturing nonlinearities and interactions under limited sample sizes (Chen & Guestrin, 2016; Ke et al., 2017; Prokhorenkova et al., 2019), and ordinal regression provides a principled, interpretable baseline when outcomes are naturally ordered (McCullagh, 2018). More recently, large language models have been explored as general-purpose predictors on structured or tabular data via serialization and prompting (e.g., TabLLM), raising the possibility that prompted inference could reduce dependence on specialized tabular pipelines (Hegselmann et al., 2022).

However, emerging evidence suggests that prompted LLMs can struggle when success depends on precise quantitative structure and sharp decision boundaries, even when they capture coarse ordering, and that adding more context can plateau or degrade accuracy rather than improve it (Cao

et al., 2026). Our work contributes at this intersection by providing a participant-level generalization benchmark on a real healthy aging EMA dataset with an ordinal cognitive severity target, directly comparing prompted LLM inference against strong tabular machine-learning and statistical ordinal baselines under the same evaluation protocol. Unlike prior LLM-on-tabular studies that often focus on generic benchmarks or do not foreground boundary-accurate ordinal decisions, we explicitly distinguish proxy-inclusive reconstruction from proxy-reduced prediction to reduce conflation with label leakage, and we treat machine-parseable structured-output compliance as part of model performance by reporting schema and parse failures as deployment-relevant outcomes rather than silently discarding them.

## 5 MEANINGFULNESS STATEMENT

Population aging has made cognitive health a central public-health concern, with the prevalence of Alzheimer's disease and related dementias (ADRD) and associated care costs continuing to rise. This motivates scalable monitoring approaches that can detect changes earlier and track day-to-day variability more faithfully than infrequent clinic visits. Intensive longitudinal designs (e.g., measurement burst) were developed precisely to separate stable between-person differences from dynamic within-person fluctuations, which are increasingly recognized as informative targets rather than mere noise (Bolger & Laurenceau, 2013).

The research gap we aim to fill is a methods-and-evaluation gap at the intersection of ambulatory cognition and foundation models. Despite growing use of smartphone EMA and cognitive micro-tests for older adults (and evidence they can be reliable and feasible in at-risk aging cohorts), the field lacks a rigorous, head-to-head, participant-level generalization benchmark that tests whether prompted LLMs can actually replace specialized tabular or ordinal learners on the kind of data these studies produce—namely, small-N, heterogeneous, predominantly tabular feature sets with ordinal clinical targets. Existing LLM-on-tabular work often evaluates on generic tabular benchmarks or settings where feature semantics are rich and sample sizes are larger, and it typically does not foreground boundary-accurate ordinal decisions nor deployment-critical parseability as first-class outcomes (Hegselmann et al., 2022; McCullagh, 2018). Meanwhile, digital health machine learning has repeatedly shown that record-wise splits can inflate apparent performance via identity confounding, making participant-level splits essential for believable claims of generalization to unseen individuals (Neto et al., 2017; Saeb et al., 2017).

## 6 CONCLUSION

We presented a systematic comparison of state-of-the-art LLMs, strong tabular machine-learning models, and classical statistical ordinal baselines for predicting an ordinal cognitive severity label from a real-world healthy aging EMA dataset. By grounding evaluation in participant-level splits, our study targets the deployment-relevant setting of generalization to unseen individuals, rather than repeated measurements from the same participants. Across models and configurations, the results draw a clear boundary between what is currently reliable and what remains aspirational: tree-based tabular models provide the strongest boundary-accurate septile predictions, reflecting their advantage in learning sharp decision thresholds from heterogeneous numeric features under small-sample conditions, while prompted LLMs tend to capture coarse ordinal ordering but fall short on exact septile decisions. Importantly, we show that operational reliability is not secondary in LLM-based pipelines—structured-output failures and schema noncompliance directly reduce usable coverage and must be included as part of end-to-end performance. Taken together, our findings argue for a pragmatic stance in ambulatory cognition modeling today: specialized tabular predictors should remain the default choice when precise ordinal decisions are required, while LLMs are better viewed as complementary components that may add value through coarse severity reasoning, flexible interfaces, or hybrid designs that couple LLM robustness with tabular precision. We hope this benchmark and analysis help the community evaluate LLM claims in small, structured clinical monitoring settings and motivate future work on reliability-aware hybrid modeling, proxy-free cognition inference, and longer-horizon longitudinal validation.

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
