# OpenReview forum: "How Far Can LLMs Go On Cognitive Health Prediction? A Study On EMA Data"
_ICLR.cc/2026/Workshop/LMRL — ICLR 2026 Workshop LMRL Poster_

### Official Review · Reviewer_7o1F · 2026-02-13
**A practical study on LLM utility for tabular EMA data with notable scaling limitations.**

**Rating:** 6
**Confidence:** 4

**Review:**

This study evaluates the effectiveness of LLMs in predicting cognitive health states using Ecological Momentary Assessment data, comparing them to traditional ML models in a structured, tabular regime.
The paper provides a timely evaluation of LLMs in a domain (clinical cognitive health) where data is notoriously heterogeneous and sparse. The authors correctly identify that while LLMs show promise in zero-shot reasoning, they struggle to outperform "shallow" ensemble methods (like XGBoost) on purely tabular EMA datasets. The clarity of the experimental setup is excellent, and the focus on EMA which captures real-world variability is highly significant. However, for a full 8-page paper, the "originality" is somewhat limited; the finding that "LLMs are not yet better than Gradient Boosting for tabular data" is well-documented in other fields. The paper would be significantly strengthened by exploring why LLMs fail here specifically, whether it is a tokenization issue or a failure to model long-term temporal dependencies in cognitive micro-tests.

Strengths: Thorough benchmarking on a difficult clinical dataset; clear articulation of LLM limitations in structured data regimes.
Potential gaps to be addressed: Limited technical novelty; lacks a proposed architectural fix for the identified LLM weaknesses; findings align closely with existing "LLM vs. Tabular" literature.

---

### Official Review · Reviewer_SSpo · 2026-02-19
**A clean evaluation of LLMs on an important real-world heathcare task**

**Rating:** 6
**Confidence:** 3

**Review:**

The authors evaluate modern LLM-based approaches against standard ML baselines on the task of detecting cognitive variability in daily life via repeated "ecological momentary assessments (EMAs)". They evaluated their models on a dataset of 115 older adults. For LLMs, they evaluated zero-shot and few (0,7,8,12) reasoning and the effect of context length, and for all tasks evaluated accuracy across on an ordered septile outcome range (0-7).

The results are clear, although I would have liked more details on the exact outcome metrics used (e.g., I am making an educated guess as to what exactly "ToleranceAccuracy@0" means).

In general, the results clearly show that standard tree-based methods on these problems essentially solve it, rendering the use of substantially more-expensive LLMs (probably) wasteful for this application. They show the expected degradation of LLM performance as context fills up (exact numbers for low, medium, and high context filling would be useful here), and show that with few-shot prompting the LLMs can pull ahead of the baseline models by the few remaining percentage points available.

In general I would have liked more methodological details - the paper as it stands is not particularly reproducible, but I think it will be of interest to healthcare domain experts and will produce meaningful discussion, so I vote to accept.

---

### Official Review · Reviewer_rRpw · 2026-02-23

**Rating:** 6
**Confidence:** 4

**Review:**

The authors present an evaluation of predicting cognition using EMA, focusing on the model comparison between classical tabular approaches and LLMs. The latter are evaluated under different levels of problem context. Results indicate that tree-based models perform best at predicting the septile scores.

Pros:
1. The proposed research question is well addressed with a relatively large model set, metrics, and evaluation scheme.
2. No clear biases in favour of LLMs as authors fairly reconsider failures to output appropriately structured responses.
3. The authors evaluate on a novel, clinical dataset, which is relevant for the workshop.

Cons:
1. A description of variability in the obtained scores is missing. This make interpretations ('model x is best') hard to assess and is especially important in these small sample sizes, where variability resulting from randomized train/test assignments can be significant.
2. The paper is fairly superficial and narrow. While a comparison across 'context levels' for the LLMs is interesting, it's also not further investigated (e.g. the authors interpret context as 'calibration rather than knowledge acquisition' but I didn't see any analysis supporting this).
3. Given 2. - the paper is too verbose and makes the same points repeatedly.
4. At least for me, it is unclear why LLMs would even be a good candidate for this kind of tabular prediction. This motivation is sorely lacking in the introduction, as the paper's motivation depends on it.

Overall, while superficial and limited, the analysis seems well executed within its narrow scope and is clearly written up.

---

### Meta-Review · Area_Chair_W54J · 2026-02-28

**Recommendation:** Accept (Poster)
**Confidence:** 4

**Metareview:**

While there were some valid concerns on the evaluation & reproducibility, the reviewers all felt that this is worth discussing at LMLR.

---

### Decision · Program_Chairs · 2026-03-02

**Decision:**

Accept (Poster)

**Comment:**

Please see the meta-review.